# Aesthetic Results, Functional Outcome and Radiographic Analysis in THA by Direct Anterior, Bikini and Postero-Lateral Approach: Is It Worth the Hassle?

**DOI:** 10.3390/jcm12031072

**Published:** 2023-01-30

**Authors:** Alberto Di Martino, Matteo Brunello, Valentino Rossomando, Davide Pederiva, Francesco Schilardi, Niccolò Stefanini, Giuseppe Geraci, Cesare Faldini

**Affiliations:** 1Ist Orthopaedic Department, IRCCS—Istituto Ortopedico Rizzoli, Via Giulio Cesare Pupilli, 1, 40136 Bologna, Italy; 2Department of Biomedical and Neuromotor Science-DIBINEM, University of Bologna, 40126 Bologna, Italy

**Keywords:** total hip arthroplasty, hip, direct anterior approach, bikini, aesthetic

## Abstract

Total hip arthroplasty (THA) can be performed by several approaches such as direct anterior (DAA), direct lateral (DL) and postero-lateral (PL). Our study was conducted to compare among different approaches, such as DAA, bikini (BK) and PL, the aesthetic impact of the scar, differences in the position of prosthetic components and differences in functional rehabilitation outcomes. Materials and methods: Population, composed by 240 patients, was collected among patients treated for primary total hip arthroplasty (THA) from 1 January 2017 to 31 December 2021 and divided by surgical approach. Of these, 160 female patients were included in the current analysis, leaving 58 DAA, 52 BK patients and 50 PL patients. Demographic and clinical parameters were retrospectively collected: age, BMI, time of surgery, length of stay, Harris Hip Score (HHS) before and after surgery at 6 months and patient, intra/post-surgical complications and Patient and Observer Scar Assessment Scale (POSAS). Results and Discussion: Our results showed a better aesthetical result in BK group compared to DAA group and faster rehabilitation with the DAA compared to PL. Optimal cup positioning was reached both in PL approach and DAA approach. DAA showed no increase in complications compared to PL approach and offered a faster recovery. Bikini approach is an alternative to the standard DAA approach and can be proposed for patients where a better aesthetic result is desired in addition to better functional recover.

## 1. Introduction

Total hip arthroplasty (THA) is one of the most successful major surgeries in orthopedics; it is safe and reliable independently, and it is currently performed by different surgical approaches such as direct anterior, antero-lateral, direct lateral, and postero-lateral (respectively, DAA, AL, DL, PL) depending on the surgeon’s preference and experience [1,2]. In the last few years, a tendency towards a decrease in invasiveness in THA surgery has been observed, which has brought on the relative increase in the performance of the DAA and PL THA procedures [3,4]. DAA has gained popularity thanks to its reduced invasiveness, given by the preservation of muscle insertions and soft tissues, which promotes early rehabilitation. The minimally invasive PL approach only requires the section of the external rotators of the hip joint and is associated with early return to function. Both preserve the abductor mechanism by leaving intact the gluteus medium muscle, the main effector of the hip joint [5,6]. Independently from the return to function, which in most patients is guaranteed by any of the abovementioned minimally invasive surgical approaches, hip arthroplasty surgery has evolved also from an aesthetic point of view. DAA can be performed through an incision that closely follows Langer’s skin lines by a transverse cutaneous wound, called the bikini (BK) technique [7,8]. It is usually reserved for young women requiring THA surgery [9]. PL has the advantage that the incision is posterior to the greater trochanter, and it is not usually easily visible by the patient when facing a mirror. However, only a few studies analyzed patients’ satisfaction in terms of the cosmetic outcome of THA wounds so far. Moreover, the minimally invasive PL and BK technique require a dedicated skill to perform the same arthroplasty surgery through a reduced and modified incision; in fact, the reduced surgical exposure can compromise the ability of the surgeon to correctly position the implants, as witnessed by studies where the minimally invasive incision may hamper the correct position of the surgical tools, increasing surgical-related risks [1]. This retrospective study was, therefore, carried out to evaluate in a female surgical population: (1) the aesthetic impact of the BK scar as opposed to the standard DAA and PL approach; (2) any radiographic difference in implant position when the BK approach is used compared with the standard DAA and PL approach; (3) any difference in functional and rehabilitation outcomes in patients operated on by BK, DAA and PL minimally invasive THA.

## 2. Materials and Methods

The following study is retrospective and this study, such as all the case collections, have been approved by the Local Ethics Committee (CE-AVEC) with the code 021 ANT-HIP, 347/2021/Oss/IOR. Population of the study has been collected among patients treated for primary THA from 1 January 2017 to 31 December 2021. Patients’ retrieval was performed through the institutional informatic database, screened for ICD-9 codes for total hip arthroplasty. Patients’ charts were then searched for the clinical, radiographic and surgical information. Inclusion criteria were: any primary total hip replacement surgery performed by the PL, standard DAA or BK approach. The exclusion criteria were: revision THA surgeries, THAs performed for a diagnosis of fracture and in patients with a previous diagnosis of infection or tumor and patients performing simultaneous bilateral THAs.

A population of 240 patients was selected, and they were divided in three groups according to the surgical approach in direct anterior (DAA), postero-lateral (PL), bikini (BK): 93 patients were in the DAA group, 94 PL and 53 BK; of these, 160 female patients were included in the current analysis, leaving 58 DAA patients, 52 BK and 50 PL (Figure 1).

DAA is performed with an approximately 6 cm incision starting 2 cm lateral and distal from the ASIS. The fascia is then incised longitudinally and slightly laterally to avoid damage on the femoral cutaneous nerve. The sartorius and tensor fascia lata muscles are then spread apart. The major ascending branch of the lateral circumflex artery is ligated. The capsulotomy is performed and then neck’s osteotomy allows to remove the femoral head.

In the BK approach, the incision is performed with an approximately 6 cm incision, following the lines of Langer, on the groin fold. The mark for the correct incision is an imaginary line perpendicular to the ASIS; thus, making the incision two thirds laterally and one third medially to the line. Using two retractors, the skin is then rotated 90°, continuing the approach in a manner similar to the anterior approach.

In PL approach, the incision of the skin begins at the level of the PSIS and extends 5 to 8 cm cephalad to the posterior portion of the greater trochanter. The incision is curved distally along the femoral shaft. The fascia is incised and the fibers of the gluteus maximus muscle are split. The hip is internally rotated and stretched to relax the gluteus maximus and expose the external rotators. To preserve the sciatic nerve while exposing the hip capsule, the short rotators can be separated at their femoral insertion and reflected backward. To retain a piece of the capsule for eventual reconstitution, a longitudinal or T-shaped capsulotomy is done. Dislocating the hip posteriorly in flexion, adduction and internal rotation exposes the femoral head and neck. If dislocation is problematic, the femoral head and neck can be osteotomized in place.

At suture, an intradermal absorbable suture was performed in 105/160 patients, in 37/58 (63.8%) DAA, in 36/52 (69%) BK and in 32/50 (64%) PL patients, respectively. All the other patients were subjected to mechanical suture with metallic staples. Sutures were performed by the residents in the operating room (Figure 2).

Patients’ demographics and clinical parameters were retrospectively collected from the medical records of the hospital: age at surgery, BMI, surgical time, in-hospital length of stay and intra and postoperative complications. Pre and 6-month postoperative Harris hip scores (HHS) [10] were recorded, together with a patient and observer scar assessment scale (POSAS) score [11]. All surgeries were performed by the hip surgery team consisting of 4 senior surgeons.

### 2.1. The Patient and Observer Scar Assessment Scale (POSAS)

The POSAS is a scar assessment score consisting of two sections: one completed by the observer and one completed by the patient. The observer scale includes different items, including vascularity and pigmentation of the scar; relief or the irregularity of the scar surface; and scar thickness, pliability, and surface area. Patient scales had a questionnaire consisting of 7 questions referring to scar pain, itching, difference in color compared to the skin, consistency to the touch, profile compared to the skin, regularity and global judgement. For each question, the patient gave a score from 1 to 10, 1 being maximum satisfaction obtained with 1 point per item, while the worst score was 10; therefore, the overall highest satisfaction was obtained with 7 total points, and the lowest with a total of 70 points. For the current study, only the patient section was used (patient scar assessment scale, PSAS), through interviews performed at follow-up.

### 2.2. Radiological Evaluation

Radiological evaluation was performed on postoperative radiographs and evaluated cup inclination and stem alignment. Radiographic analysis was performed by 3 independent hip surgeons to keep systematic error rate low. Cup inclination was measured as the angle between the line tangent to the border of the acetabular cup and a line parallel to the horizontal plane passing through the drop sign or ischial tuberosity and was compared to Lewinnek’s safe zones, 40° ± 10. Stem alignment was measured as the angle between the femoral axis and stem axis; alignment was considered good when the stem axis was parallel to the femoral axis with a tolerance of 5° in varus or in valgus (−5° and +5°, respectively).

### 2.3. Statistical Analysis

The analyses were conducted comparing the groups with t-student test; significance was set at *p*-value < 0.05. (SPSS 14.0, version 14.0.1 (SPSS Inc., Chicago, IL, USA).

## 3. Results

### 3.1. Patients’ Demographics and Characteristics

Age at surgery in the DAA group averaged 59.8 years (range, 25–84 years). The BK group presented an average age of 56.9 years (range, 24–75 years). The PL group presented an average age of 60.2 years (range, 24–82 years), not significantly differing from the DAA (*p* = 0.11 T Student) and BK patients (*p* = 0.066 T Student). The BMI was lower in the BK group compared to the DAA and PL patients, but the DAA–PL and BK–PL comparison did not show any significantly difference (*p* = 0.063). In the BK group, the average BMI was 24.5 ± 2.7, in the DAA group, the average BMI was 25.6 ± 5.32, while in PL group, the average BMI was 27 ± 4.1.

Surgical time showed not differences between the groups (*p* = 0.49). The DAA group had an average surgical time of 87.76 ± 22.1 min; the BK group had 84.3 ± 12.8 min while the PL group registered an average of 87.37 ± 22.1 min. Hospitalization time did not show differences in the three matched groups. The DAA group had an average hospitalization of 10 ± 3 days, the BK group had an average days of 8.5 ± 1.5 and the PL group reached an average of 10 ± 2 days (DAA and BK: *p* = 0.039; BK and PL: *p* = 0.034; DAA and PL: *p* = 0.73 T Student).

Functional outcome evaluated by HHS in pre- and post-surgery did not show a significant difference between the matched groups. The DAA and PL groups had pre-intervention HHS (49.9 ± 17.3 and 52.3 ± 15, respectively) (*p* = 0.722) and post-intervention HHS (92.8 ± 7.48 and 92.7 ± 5.2, respectively) (*p* = 0.43). The DAA and BK groups had pre-surgery HHS (49.9 ± 17.3 and 48.9 ± 9) (*p* = 0.2) and post-surgery HHS (92.8 ± 7.48 and 92.8 ± 9.53) (*p* = 0.72). The BK and PL groups had a pre-surgery HHS intervention HHS (48.9 ± 9 and 52.3 ± 15, respectively) (*p* = 0.36) and post-surgery HHS (92.8 ± 9.53 and 92.7 ± 5.2, respectively) (*p* = 0.39) (Table 1).

An overall rate of complications of 4.37% (7/160) was observed. In the DAA group, 1 dislocation was observed, treated by open reduction. In the BK group, 1 aseptic loosening of the acetabular component treated by cup revision and 3 femoral-cutaneous nerve paraesthesia were recorded, 2 of which resolved after 6 months and one in 12 months. The PL group reported 1 intraoperative femur fracture, managed by metal wiring. The rate of complications did not show significant differences among groups (Table 2).

Esthetical outcomes evaluated by the patient POSAS questionnaire showed significantly better esthetic results in the BK group (11.7 ± 4) compared to the DAA group (14.1 ± 3.9) (*p* = 0.021). The POSAS questionnaire of the PL group showed an average of (14.5 ± 3), without statistical difference compared to the DAA group (*p* = 0.62), but significantly more than the BK patients (*p* = 0.0032) (Table 3).

### 3.2. Radiographic Analysis

Cup inclination analysis showed that BK patients had cup inclination of 38.4° ± 4.2°, closer to the Lewinnek safe zone of 40° when compared to DAA patients who had 35.3° ± 7.1° (*p* = 0.042), and close to the PL group with an average 37.8° ± 9.6 (*p* = 0.07). Overall, all the patients on average showed cup placement in the Lewinnek safe zone (40° ± 10°) (Table 4).

## 4. Discussion

The purpose of this study was to evaluate the aesthetic result of the surgical wounds and patients’ satisfaction with the surgical incision, together with functional and radiographic outcomes of THAs performed by two minimally invasive approaches with three surgical incisions: anterior or longitudinal, bikini and postero-lateral. We found that each approach allowed optimal functional results, with cup placement closer to Lewinnek’s safe zone for the BK group, and better aesthetical results in the BK group compared to the DAA and PL groups.

The comparison between the standard DAA and bikini DAA in terms of aesthetic outcome is debated in the recent literature. Leunig et al. [12] compared scar satisfaction in standard longitudinal to oblique “BK” incision in THAs performed by DAA and observed a higher satisfaction rate with the BK incision compared to the longitudinal incision, without differences in complications; they solely questioned the patients about the overall satisfaction of the aesthetic outcome. In his following study in 2018, Leunig et al. [13] compared a larger population of 964 patients who performed THA with BK and DAA, evaluating aesthetical outcomes by the UNC4P-a score, which assesses itching, pain, paresthesia and scar texture. They confirmed better results for the bikini approach when compared to the standard DAA. Moreover, in 2021, Wang Q et al. [14] compared cosmetic satisfaction in the bikini approach compared to DAA, observing greater patients’ satisfaction in cosmetic aspects of their scars compared with patients who received traditional longitudinal incisions. In contrast, in their study, Manrique et al. [15] indagated the difference in BK and DAA in obese patients in terms of satisfaction of the scar; they did not observe any difference between the two approaches using two questionnaires (6-parameter POSAS and Vancouver Scar Scale).

Despite the scanty literature present on the topic, the results obtained from the current study confirms a better patient satisfaction with the BK incision compared to the standard longitudinal DAA. In our study, a higher complication rate in absolute terms was shown in the BK group compared to the DAA group, despite the former had a lower average BMI. From our point of view, the complication defined as neuroapraxia has less ‘‘weight’’ compared with other “major” complications including intraoperative fracture or dislocation, so we urge a critical judgment of Table 2 as opposed to a mere view of the raw data.

We consider dislocation, loosening and fracture as major complications in that they require more hospitalization, in the case of intraoperative fracture, or even a second hospitalization as in the case of loosening or dislocation.

Fundamental to take into account are the consequences for the patient who must be subjected to, in addition to considerable emotional distress, further hospitalization with all the risks related to a second hospitalization and a second surgery [16].

Moreover, from an economic point of view, as evidenced by several studies [17,18], intraoperative periprosthetic fractures, dislocations and mobilizations have a significant impact on hospital systems. In fact, the aforementioned complications are burdened with high cost, in particular when revision implants are required [19].

To date, there are no studies comparing the BK approach with the PL one. The analysis of our data showed suggest a potential advantage of the BK compared to the PL approach, in terms of overall satisfaction.

Radiological findings evidenced that all accesses allowed cup implant placement with an inclination close to the Lewinnek safe zone of 40°. Radiological parameters are easier and faster methods to calculate cup inclination, theorized from Lewinnek in 1978 [20] who defined the safety parameters for cup placement: a cup tilt at 40° ± 10° and an anteversion at 15° ± 10°, were related with a lower rate of dislocation. Although there are many studies that refute these parameters [21], at present, the cup is implanted according to these guidelines. As regards the influence of surgical approach on cup positioning and orientation, Leunig et al. [12,13], in their two comparative studies between the BK and DAA published in 2013 and 2018, did not observe differences in implant orientation on radiographic exams.

The results of current study, contrarily to our baseline hypothesis, showed a cup placement closer to Lewinnek’s safe parameters when performed by the BK approach (38.4 ± 4.2°) compared with the standard DAA incision (35.3 ± 7.1°); however, both values were within the “safe zone”. The greater accuracy in cup placement observed in this study, using the BK access could be related to the fact that it is mostly performed in younger female patients, with reduced BMI [9].

Regarding function, general agreement is observed in the literature on long-term recovery when comparing the anterior accesses, DAA or BK, with the PL approach. Several authors, including Putananon et al. [1] and Jia et al. [22], compared the DAA and PL approaches in THA and found a great functional improvement in both approaches, using HHS before and after surgery. Similar conclusions were obtained by Rodriguez et al. [23] who compared the DAA and conventional PL approach, evaluating functional scores such as the Timed Up and Go (TUG) test, the motor component of the Functional Independence Measure TM (M-FIMTM), the UCLA activity score, and the HHS. They observed optimal functional results when either of the surgical approaches were used. This result emphasizes the faster functional recovery in the immediate postoperative period due to the muscle sparing that is observed in THAs performed by DAA compared with PL.

The main limitations of the study are represented by the difference in demographic and numerosity of the three groups (BK, DAA and PL) and the variable interval between intervention and interview. In particular, the BK approach, being proposed more to young women, is still performed less frequently than the standard DAA and PL; for this reason, we evaluated a selected population of female patients to address the study questions in a more uniform fashion. The same study limitations were found by Leuning et al. [12,13] in whose study, the BK approach was performed predominantly in young slim women. We are aware of the limitation of the study, given the brevity of the follow-up, and it might be interesting to investigate a prospective population with long-term results in the future, particularly with regard to the long-term modifications of the HHS.

## 5. Conclusions

In conclusion, we can affirm that in a selected female population we found a more rapid post-operative recovery of THA with the DAA or BK approaches compared to the PL one; all these approaches allowed an optimal cup positioning. Regarding the aesthetic outcomes, BK incision resulted in better satisfaction rate compared to standard DAA; comparison with PL approach did not show any significant difference, even though there was a trend towards greater overall satisfaction in patients operated by the BK approach. The anterior approach therefore represents an excellent option for the performance of THA surgery, showing an acceptable complication rate risk guaranteeing a faster recovery; moreover, in selected patients, the BK approach results support a valuable alternative to the standard DAA, and it can be proposed for female patients who desired, in addition to functional recovery, a better aesthetic result.

## Figures and Tables

**Figure 1 jcm-12-01072-f001:**
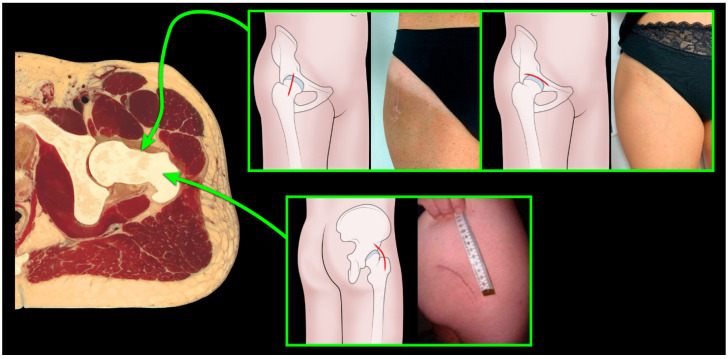
Pelvic transverse section with anterior and posterior approach, divided to incision: Bikini, direct anterior and postero-lateral.

**Figure 2 jcm-12-01072-f002:**
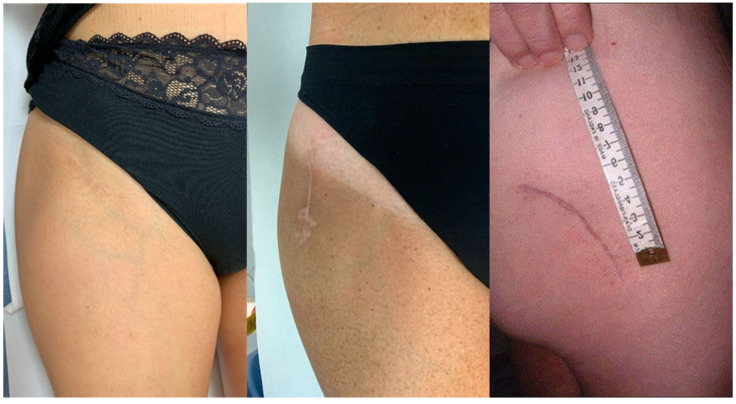
Clinical scar outcomes with three different approaches.

**Table 1 jcm-12-01072-t001:** Patient’s demographic, surgical and functional characteristics.

	DAA	BK	PL
Number of patients	58	52	50
Age at surgery (years)	59.8	56.9	60.2
BMI	25.6 ± 5.32	24.5 ± 2.7	27 ± 4.1
Surgical Time (minutes)	87.76 ± 22.1	84.3 ± 12.8	87.37 ± 22.1
Intradermal suture	63.8%	69%	64%
Hospitalization time (days)	10 ± 3	8.5 ± 1.5	10 ± 2
HHS pre-surgery	49.9 ± 17.3	48.9 ± 9	52.3 ± 15
HHS post-surgery	92.8 ± 7.48	92.8 ± 9.53	92.7 ± 5.2

**Table 2 jcm-12-01072-t002:** Complications divided by approach and incision.

Complications	DAA	BK	PL	Total
Femoral-cutaneous nerve apraxia	-	3/52 (5.77%)	-	3/160 (1.87%)
Intraoperative femur fracture	-	-	1/50 (2%)	1/160 (0.62%)
Dislocation	1/58 (1.7%)	-	-	1/160 (0.62%)
Aseptic mobilization	-	1/52 (1.92%)	-	1/160 (0.62%)
Total	2/93 (1.7%)	4/52 (7.69%)	1/50 (2%)	7/160 (4.37%)

**Table 3 jcm-12-01072-t003:** POSAS patient scale results divided by approach.

POSAS Patient Scale Results	DAA	BK	PL
Pain	1.2	1.9	1.5
Itching	1.3	2.4	1.5
Colour	3.3	1.9	2.5
Stiffness	2.2	1.3	2.4
Thickness	2.5	1.4	2.5
Irregular	1.5	1.5	1.5
Opinion	2.1	1.3	2.6
Total	14.1	11.7	14.5

**Table 4 jcm-12-01072-t004:** Implant placement with the cup inclination, divided by approach and incision.

Cup Inclination	DAA	BK	PL
Mean	35.3	38.4	37.8
Standard deviation	7.1	4.2	9.6
Minimum	25	27	23
Maximum	49	48	61

## Data Availability

All collected data are reported in the current manuscript.

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
