# Peer review of "Aesthetic Results, Functional Outcome and Radiographic Analysis in THA by Direct Anterior, Bikini and Postero-Lateral Approach: Is It Worth the Hassle?"

_jcm, 2023, doi:10.3390/jcm12031072_

Round 1
Reviewer 1 Report
Major
1. This study evaluated the comparison among standard direct anterior (DA), bikini (BK) and postero-lateral (PL) THA approach. The first sentence may be better to change.
2. Authors described that each approach allowed to optimal functional results, with a closer cup placement to Lewinnek’s safe zone for BK group, and better aesthetical results in BK group compared to AD group. However, aesthetical results were shown only POSAS questionnaire score. Authors should show observer scale (Vascularity, Pigmentation of the scar, Relief or the irregularity of the scar surface, scar Thickness, Pliability, and Surface area) and patient scales (scar pain, itching, difference in color compared to the skin, consistency to the touch, profile compared to the skin, regularity, and global judgement) in detail as Table.
3. Although BMI was significantly lower than the direct anterior (DA) group, the bikini (BK) group had the highest frequency of complications. From this I don't think the BK group is superior. Authors should discuss the complications in the BK group
4. This paper has many mistakes in abbreviations. Several different abbreviations are used. Direct anterior are DAA, DA and AD. Bikini are BK and B.
Author Response
we tried to address all the reviewer comments. hopefully now the manuscript is suitable for publication

Reviewer 2 Report
Thank you for the opportunity to review. I have some specific remarks as following.
1. Six months after surgery is too short to assess hip joint function. From the results of this study, it is difficult to refer to the bikini cut and other approaches from the viewpoint of hip joint function.
2. Was this study all done by a single surgeon? It is desirable to describe how many surgeons are involved in the operation of this research.
3. It is necessary to describe how each approach is used properly.
4. Demographic data has many items with significant differences, so I think it would be more helpful for readers to create a table. In addition, a table should be created for at least the items listed as the purpose of this research.
5. Regarding cup installation, was there no difference in the installation rate within the safe zone?
6. Please include the results of the PL group's POSAS questionnaire in the text. Or create a table and let the reader know the results.
7. At what point in time is evaluation performed for lateral femoral cutaneous neuropathy? Is the incidence so low even in the early postoperative period?
8. It would be better to describe the statistical results in a table.
9. The conclusion of this study is that the B group did not show any significant difference compared to the PL group. It would not be appropriate to include only a comparison with the DAA group in the conclusion. From the results of this study, I feel that the significance of doing the bikini approach cannot be found...
The problem with this study is that the follow-up period is as short as half a year, and there are differences in patient backgrounds in each group. At least, it is desirable to adjust the patient background by propensity score matching. There is a possibility that bias exists in the analyzed cases, and the validity of the results obtained from this study cannot be found.
Author Response
We tried to address all the reviewer comments. We trust that the manuscript now is improved according to the reviewer's standards.
Regards

Reviewer 3 Report
Dear authors,
The title of your manuscript start with "Aesthetic results THA...". That suggests that the main purpose of your work is to evaluate aesthetic outcome of a different approaches. Later in text, you evaluate position of the implant and rehabilitation outcome also. It is very important to notice that because, in my opinion, you didn't prove a thesis that aesthetic outcome is better with bikini incision, what obviously was your primary intention (I'll explain later). I recommend a changes in manuscript title, the title should match the result and conclusion.
Because you used a data from databases, your study is retrospective. I recommend te ephasy this in chapter Introduction and Material and Methods
Inside the text, on few places I found different signs for different approaches (examples, for bikini incision B and BK, for direct anterior approach AD and DA). Please, be sure that you use a same and only one sign for any approach.
In line 33 brackets are on an inappropriate places. It is enough to write all approaches in one brackets.
Figures in your manuscript should be larger and in better resolution (especially Figure 1)
In subchapter Radiological evaluation you should clearly explain who analyzed and evaluated postoperative radiographs (radiologist, orthopaedic surgeon). You also should write how many person analyzed it. Usualy more person are involved in purpose of prevention of bias and mistake. In case that only one person analyzed and evaluated all radiographs, it should do it three times and final result should be aritmetical mean.
One of the purposes of this work is comparison in aesthetical outcome of a different surgical approaches in THA. Because of that, you should detailed explain (in chapter Materials and Methods) what kind of suture material you used, what kind of suture you used (single or returned, extended etc.) and also, who did sutures (the main surgeon or assistant). This opservation is very important because you can't compare different things.
In line 165 you made a mistake in names of approaches. It should be: anterior or longitudinal, bikini and posterolateral.
Reference 3 is written in an inappropriate way.
The main complaint I have about your manuscript is a study design. From chapter Results is obvious that in bikini group gender distribution is completely different than in other two groups. In bikini group are female dominantly (only one male patient), what is huge difference in comparison with other two groups. BMI is significantly lower in bikini group also. Because of these facts, I don't agree with your conclusion in line 232-235. You can't say that regarding aesthetical outcome satisfaction of patients from bikini group is better. Maybe, but only for female patients because you don't have male patients in this group. My recommendation is to repeat analysis of your data. Because of lack of male patients in bikini group, you should redesign you study in way to analyse independently male and female patients. After that, you will be able to give two different conclusions, divided for female and male patients. But, you won't be able to say that satisfactions of male patients operated with the bikini approach is higher or lower, because you have only one male patient in this group.
I hope my comments will help you to improve your manuscript.
Author Response
we tried to reply to the comments, and extensively re-wrote the manuscript.
hopefully now it fits the standard for publication in JCM

Round 2
Reviewer 1 Report
Thank you for replying to my comments.
Revision is acceptable.
Author Response
We would thank the reviewer for the advices, to allow improving of the manuscript
Reviewer 2 Report
In general, I think that the manuscript has been corrected.
Author Response

(The authors gave the same response as above.)

Reviewer 3 Report
I'm satisfied with your answers on my previous complaints. I have one more suggestion. You should correct a total number of patients where intradermal suture was performed. In your reply is a number of 105 patients (what is correct number obviously). In a new version of the manuscript is a number of 99 patients. I don't have any additional complaints.
Author Response
We would thank the reviewer for the advices, to allow improving of the manuscript.
We provide to clarify the following point in line 106.